# [^99m^Tc]Tc-PSMA-T4—Novel SPECT Tracer for Metastatic PCa: From Bench to Clinic

**DOI:** 10.3390/molecules27217216

**Published:** 2022-10-25

**Authors:** Michał Maurin, Monika Wyczółkowska, Agnieszka Sawicka, Arkadiusz Eugeniusz Sikora, Urszula Karczmarczyk, Barbara Janota, Marcin Radzik, Dominik Kłudkiewicz, Justyna Pijarowska-Kruszyna, Antoni Jaroń, Wioletta Wojdowska, Piotr Garnuszek

**Affiliations:** National Centre for Nuclear Research, Radioisotope Centre POLATOM, 05-400 Otwock, Poland

**Keywords:** PSMA-HYNIC ligands, PSMA-T4, kit formulation development, preclinical study, in vitro, in vivo

## Abstract

Despite significant advances in nuclear medicine for diagnosing and treating prostate cancer (PCa), research into new ligands with increasingly better biological properties is still ongoing. Prostate-specific membrane antigen (PSMA) ligands show great potential as radioisotope carriers for the diagnosis and therapy of patients with metastatic PCa. PSMA is expressed in most types of prostate cancer, and its expression is increased in poorly differentiated, metastatic, and hormone-refractory cancers; therefore, it may be a valuable target for the development of radiopharmaceuticals and radioligands, such as urea PSMA inhibitors, for the precise diagnosis, staging, and treatment of prostate cancer. Four developed PSMA-HYNIC inhibitors for technetium-99m labeling and subsequent diagnosis were subjected to preclinical in vitro and in vivo studies to evaluate and compare their diagnostic properties. Among the studied compounds, the PSMA-T4 (Glu-CO-Lys-L-Trp-4-Amc-HYNIC) inhibitor showed the best biological properties for the diagnosis of PCa metastases. [^99m^Tc]Tc-PSMA-T4 also showed effectiveness in single-photon emission computed tomography (SPECT) studies in humans, and soon, its usefulness will be extensively evaluated in phase 2/3 clinical trials.

## 1. Introduction

Prostate cancer is the second most common malignancy in men, and its prognosis depends on the stage of the disease [1,2]. Many cases of PCa are curable if they are detected early. However, a number of patients will still progress to metastatic cancers that evolve towards hormone resistance or metastatic castration-resistance prostate cancer (mCRPCa) [2]. Therefore, early detection of metastases or recurrent prostate cancer is of great clinical importance in terms of clinical evaluation, prognosis, and treatment [3,4,5].

Prostate cancer has been initially diagnosed by testing the prostate-specific antigen (PSA) level in the blood, through digital rectal examination (DRE) or prostate gland biopsy tests. However, measurement of PSA level is related to false-positive results caused by diseases other than prostate cancer, such as benign prostatic hyperplasia (BPH), urological manipulations, or prostatitis, and the biopsy is associated with numerous complications such as risks of bleeding and infection [5,6]. Furthermore, currently, morphological imaging such as ultrasound (US), computerized tomography (CT), magnetic resonance imaging (MRI), and metabolic positron emission tomography—computed tomography (PET/CT) imaging with [^11^C]-choline have lower sensitivity to nodal metastasis and small tumor recurrence [7,8,9].

Despite significant advances in nuclear medicine for the diagnosis and treatment of prostate cancer, research into new ligands with increasingly better biological properties is still ongoing. Prostate-specific membrane antigen is an attractive and promising target for the specific imaging and treatment of primary and metastatic prostate cancer [10,11]. PSMA ligands show great potential as radioisotope carriers for the diagnosis and therapy of patients with metastatic PCa. 

PSMA is membrane zinc metalloprotease, also known as glutamate carboxypeptidase II (GCPII), which is mainly expressed in normal human prostate epithelium and is enormously increased up to thousand-fold in PCa, including metastatic disease [10,12]. PSMA is expressed in most types of prostate cancers, and its expression is enhanced in poorly differentiated, metastatic and hormone-refractory carcinomas, and hence it can be a valuable target for developing radiopharmaceutical and radiolabeled inhibitors, such as urea-based PSMA inhibitors, for precisely diagnosing, staging, and treating PCa [13,14].

Various anti-PSMA radiopharmaceuticals were tested to develop ligands that would have great potential in preclinical and clinical stages, enabling effective diagnosis and therapy of PCa. Currently, several molecular imaging approaches give more accurate information about the disease stage and allow the selection of patients for appropriate treatment. Most of them utilize PET technology using ^68^Ga or ^18^F radioisotopes, which are conjugated with PSMA ligands [11,15,16,17,18,19]. Among the different PSMA-targeting tracers available, the most commonly used for prostate cancer PET imaging is [^68^Ga]Ga-PSMA-11, also known as gallium (^68^Ga) gozetotide, Glu-NH-CO-Lys-(ahx)-HBED-CC-[^68^Ga]Ga (HBED-CC: N,N′-bis[2-hydroxy-5-(carboxyethyl)benzyl]ethylene diamine-N,N′-diacetic acid) or [^68^Ga]Ga-PSMA-HBED-CC [15,20,21,22]. However, the time limitation associated with the use of short-lived Ga-68 and the inability of HBED-CC chelator to create complexes with other diagnostic and therapeutic radionuclides such as ^99m^Tc, ^18^F, ^111^In, ^177^L, ^90^Y, etc., highlights the need to develop new radiopharmaceuticals for PCa diagnosis and therapy [23,24,25,26,27,28,29].

Even though in recent years, the interest in PET diagnostic procedures has significantly increased, the SPECT method using radiopharmaceuticals labeled with technetium-99m (^99m^Tc) still accounts for the vast majority of nuclear medicine procedures (>85%). Technetium-99m is the most popular diagnostic radionuclide, which results from its physical properties (T_1/2_ 6.01 h, Eγ 141 keV), availability (^99^Mo/^99m^Tc generators), rich coordination chemistry, and low radiotoxicity.

Several publications concerning the PSMA inhibitors labeled with ^99m^Tc, containing a carbonyl system for radionuclide bonding in their structure, were already released over ten years ago [17,25,30,31]. However, described preparations had some disadvantages, such as slow pharmacokinetics, high liver uptake, and slow clearance from the gastrointestinal tract, which can disrupt their application in prostate cancer imaging because this type of cancer most frequently metastasizes in the lower part of the spine, pelvis, and lymph nodes within the abdomen. Therefore, the priority was to develop PSMA inhibitors labeled with ^99m^Tc of improved biodistribution and pharmacokinetics. In recent years, many PSMA inhibitor analogues containing chelating systems of N_4_ and N_3_S or 2-hydrazinonicotinic acid (HYNIC) for technetium-99m have been developed [32,33,34,35,36]. In their concise review article, Brunello et al. [37] discuss several dozen PSMA inhibitor analogues for technetium-99m labeling, including seven with the HYNIC chelating system. They briefly summarize the results of preclinical studies for ^99m^Tc-labeled PSMA inhibitors and for five, published results from clinical trials. Among the tracers discussed with the HYNIC chelating system, there is also [^99m^Tc]Tc-PSMA-T4, a preparation developed by our team [38,39]. PSMA-T4 (Glu-CO-Lys-L-Trp-4-Amc-HYNIC) as a potential carrier for technetium-99m was selected from a group of compounds in which the chain linking the Glu-CO-Lys pharmacophore system to the HYNIC chelator was modified. It was presented by Benesova et al. [26,40] that the pharmacokinetics properties, including PSMA inhibition potencies, cellular internalization, and biodistribution behavior of the PSMA inhibitors, can be significantly influenced by modification of the linker. Thus, the linker moiety’s chemical constitution significantly impacts the in vivo tumor-targeting and pharmacokinetics of PSMA-targeting radioligands. Their research resulted in developing the PSMA-617 inhibitor, which had favorable biological properties and became the gold standard as a carrier for lutetium-177 for mPCa therapy. Similar conclusions about the significant effect of linker design on the in vitro and in vivo affinity for PSMA and the change in biodistribution profile were drawn by Wirtz et al. [41], subjecting the PSMA inhibitor PSMA-I&T to a modification towards increased lipophilicity.

Therefore, in the development process of the new PSMA ligand, we focused on using the most appropriate linker system to improve the pharmacokinetics of the final ^99m^Tc-labeled preparation. The undertaken studies showed that the presence of L-Trp in PSMA-T4, as one of the linkers, instead of naphthylalanine (L-2NaI), has led to a significant improvement in the biodistribution of the labeled molecule (reducing kidney accumulation) and its increased affinity to PSMA in vivo.

This work presents the results of comparative preclinical studies for four developed small-molecule inhibitors of PSMA with modified linker systems in order to select the one with the most favorable in vitro and in vivo biological properties.

## 2. Results and Discussion

### 2.1. Synthesis

The designed PSMA-HYNIC compounds, Figure 1, were synthesized on the solid phase followed by C18 reversed phase high-performance liquid chromatography (HPLC) purification, resulting peptides of >97% HPLC purity. The identity of synthesized compounds was confirmed by mass spectrometry (MS) and, in the case of PSMA-T4, additionally by the nuclear magnetic resonance (NMR), Appendix A.

### 2.2. Radiolabeling, QC, Stability

Synthesized PSMA-HYNIC compounds were radiolabeled with ^99m^Tc in presence of tricine and N,N-diacetic acid (EDDA) as co-ligands, Appendix A. The radiochemical purity was >95% for all labeled compounds, determined via radio HPLC and thin layer chromatography (TLC) methods.

Characterization of radiochemical impurities was achieved by HPLC with the aid of suitable radioactive samples, such as Na[^99m^Tc]TcO_4_, [^99m^Tc]Tc-Tricine, [^99m^Tc]Tc-EDDA, [^99m^Tc]Tc-(EDDA)_2_-PSMA-T4. The respective retention times (Rt) were: Na[^99m^Tc]TcO_4_: Rt 4.0 min, [^99m^Tc]Tc-Tricine: Rt 2.5 min, [^99m^Tc]Tc-EDDA: Rt 2.4 min, [^99m^Tc]Tc-(EDDA)_2_-PSMA-T4: Rt 10.8 min, Figure 2.

In the radiochromatograms of all labeled PSMA-HYNIC analogs, a minor signal before the main peak was observed. The identity of this species was assessed for PSMA-T4 radiocomplex by liquid chromatography-mass spectrometry (LC-MS) method using the long-lived technetium-99. The sample was ionized using the electro-spray method and the mass was determined in positive ionization mode. The main peak on the radio-chromatogram with retention time of 10.2 min and a small peak with retention time of 9.9 min had the same mass of the molecular ion found in LC-MS at m/z 613.6 ([M+2H]^2+^) and m/z 1226.3 ([M+H]^+^). The masses observed were consistent with the addition of 1 technetium and 2 EDDA molecules to the peptide, with the concomitant displacement of 5 protons, Table 1. The presence of labeled species containing one EDDA molecule, mixed tricine/EDDA ternary complex, and an oxo or halide group in complexes was excluded. The binding of a technetium atom to PSMA-T4 and co-ligands is accompanied by the displacement of five hydrogen atoms, implicating the presence of technetium in oxidation state (V). The presence of two peaks with the same mass of ions found in the LC-MS studies, which corresponded to [^99m^Tc]Tc-(EDDA)_2_-PSMA-T4 complex, could be the result of isomerism in the technetium coordination sphere. The presence of two peaks with the same mass of ions found in the LC-MS studies, which corresponded to [^99m^Tc]Tc-(EDDA)_2_-PSMA-T4 complex (hereafter abbreviated as [^99m^Tc]Tc-PSMA-T4), could be the result of isomerism in the technetium coordination sphere. HYNIC is capable of coordinating technetium in monodentate and bidentate modes. Several possible structures can be proposed for technetium-HYNIC complexes, according to the interpretation of the structure of model technetium and rhenium complexes with hydrazinopyridine [42]. The resulting interpretation favors a bidentate chelating role for HYNIC, in which both the hydrazine and pyridine groups are coordinated with losing all free hydrazinic hydrogens and two hydrogens from the co-ligand molecules. There is no direct possibility to identify which EDDA donor atoms bind to technetium.

The stability studies in phosphate buffer saline (PBS) and human serum at 37 °C revealed that the radiolabeled compounds tested were stable for at least 4 h, Figure 3, with no statistically significant differences between compounds.

The study on the lipophilicity of the technetium-99m labeled PSMA ligands showed that due to use of tryptophan (L-Trp) in the structure of the linker instead of naphthylala-nine (L-2NaI), PSMA-T3 and -T4 are more hydrophilic than PSMA-T1 and -T2 analogues, which logD values are comparable with reference [^99m^Tc]Tc-iPSMA [35], Table 2. 

### 2.3. In Vitro Study

To determine the cell binding and internalization of four potential diagnostic PSMA-HYNIC inhibitors: [^99m^Tc]Tc-PSMA-T1, [^99m^Tc]Tc-PSMA-T2, [^99m^Tc]Tc-PSMA-T3 and [^99m^Tc]Tc-PSMA-T4 were incubated with androgen-sensitive human prostate adenocarcinoma cells (LNCaP cells). Preliminary studies have shown that each of tested PSMA analogues bind to LNCaP prostate cancer cells (cell uptake ~30%), Table 3: The internalization of all PSMA-HYNIC inhibitors tested were comparable and ranged from 7.5% to 9.7%.

Saturation binding studies demonstrated the high specificity of newly developed PSMA analogues binding to PSMA antigen expressed on prostate cancer cells, higher than 98%, Table 4. The calculated values of the dissociation equilibrium constants, K_d_, for [^99m^Tc]Tc-PSMA-T1 and [^99m^Tc]Tc-PSMA-T2 are similar to literature value for PSMA-11 (K_d_ 11.4) [15] while the [^99m^Tc]Tc-PSMA-T3 and [^99m^Tc]Tc-PSMA-T4 exhibit clearly higher affinity with two times lower K_d_ values, Table 4.

Binding affinities (IC50) values for cold ligands PSMA-T1, PSMA-T2, PSMA-T3, and PSMA-T4 were determined in the competitive binding assay using LNCaP prostate cancer cell membranes and [^131^I]I-MIP1095 as a competitive radioligand. As a gold standard, the widely used PSMA-11 inhibitor was also tested.

The study showed that all the PSMA-HYNIC analogs had a higher binding affinity to the LNCaP cells than PSMA-11 used as a reference substance (Table 5, Figure 4). The most promising of the four studied were PSMA-T3 and -T4 with IC50 in the 70–80 nM range.

### 2.4. In Vivo Study

The selection of the best of the synthesized PSMA-HYNIC compounds for pharmacokinetics and toxicity studies was based on a single-point (4 h i.v.) physiological distribution study of the four radioligands compared to the reference [^99m^Tc]Tc-iPSMA complex. The obtained results are presented in Table 6. The ordinary two-way ANOVA with Tukey’s multiple comparisons test (GraphPad Prism 9.4 for Windows [43]) was used for statistical analysis. 

As a result, we found out that the statistically significant differences (*p* < 0.05) were for kidneys uptake (*p* < 0.0001) and urine excretion (*p* < 0.0001) for all of the developed PSMA ligands comparing to [^99m^Tc]Tc-iPSMA, Appendix A. The uptake of the analyzed radioligands in other of the examined organs was non-significant (*p* > 0.05) compared to the [^99m^Tc]Tc-iPSMA uptake. In our comparative biodistribution study, we noted the lowest renal uptake (15.9 ± 2.09 %ID/g) and highest urinary excretion of radioactivity (91.23 ± 1.09 %ID) for [^99m^Tc]Tc-PSMA-T4. 

Therefore, among the studied compounds, PSMA-T4 showed the best physiological distribution thanks to low kidney accumulation and was selected for further in vivo examination.

The pharmacokinetics data of [^99m^Tc]Tc-PSMA-T4 in healthy rats are summarized in Table 7 as %ID/g (mean ± SD, *n* = 5), whereas the time-dependent clearance curve selectively for blood is presented in Figure 5.

[^99m^Tc]Tc-PSMA-T4 demonstrated rapid pharmacokinetics. The average radioactivity concentration in blood was low, showing fast clearance from blood circulation, Figure 6. The span value calculated using GraphPad Prism software was 0.9199, the K value was 1.765, and the [^99m^Tc]Tc-PSMA-T4 was distributed rapidly with half-life: 0.39 h. The very short biological half-life could be related to the structure of the linker (containing the L-tryptophan) and the high hydrophilicity of the radioactive molecule.

The primary route of [^99m^Tc]Tc-PSMA-T4 excretion is through the kidneys.

Up to 4 h, a decrease in blood activity was observed together with the uptake of the product in the kidneys.

The accumulation profile of [^99m^Tc]Tc-PSMA-T4 in kidneys is shown in Figure 6.

The span1 and span2 values are −177.6 and 186.5, respectively. The K1 and K2 values are 0.5273 and 0.2446, respectively. [^99m^Tc]Tc-PSMA-T4 was excreted via rats kidneys with the half-life 1: 1.314 h and half-life 2: 2.833 h.

This profile of renal clearance with a clear peak in radioactivity accumulation 2–4 h after administration may indicate that glomerular filtration is not the only mechanism in the excretion of [^99m^Tc]Tc-PSMA-T4. This process is probably accompanied by the reabsorption of the radiopharmaceutical.

Several research groups [44,45,46,47] have shown that variously labeled urea-based molecules have an unusually high uptake in mouse kidneys. This uptake is mainly specific as a consequence of physiologically expressed PSMA in mouse kidneys [44] but is also associated with renally localized glutamate carboxypeptidase II (NAALADase) belonging to the metallopeptidase family [48]. NAALADase has been located in both neural and non-neural tissues, such as the kidney, prostate, and small intestine [49,50,51]. Luthi-Carter et al. cloned, characterized, and found the NAALADase to be homologous to the prostate cancer marker PSMA [52]. Therefore, it is highly likely that such a specific accumulation of [^99m^Tc]Tc-PSMA-T4 in the kidneys is due to the presence of binding targets in the animals’ kidneys. Their concentration may vary depending on the type and species of rodent.

Biodistribution data of [^99m^Tc]Tc-PSMA-T4 in mice bearing LNCaP xenografts at 1 h, 2 h, 4 h, 6 h, and 24 h p.i. are presented in Table 8, as %ID/g (mean ± SD, *n* = 4).

The tumor uptake reached a value of 26.5 ± 1.4 %ID/g at 2 h p.i.v. and persisted up to 24 h. Interestingly, renal uptake of radioactivity was 132.5 ± 23.4 %ID/g, 41.51 ± 5.6 %ID/g and 7.16 ± 3.2 %ID/g at 2 h, 6 h and 24 h, respectively, with the fast excretion into the urine.

Relatively high tracer concentrations were observed in the spleen, in which the %ID/g was more than 1% up to 6 h (4.1 ± 0.9 %ID/g and 1.5 ± 0.5 %ID/g at 2 h and 6 h, respectively).

Specific binding of PSMA-T4 to PSMA receptors in the LNCaP cells was confirmed by the in vivo study using tumor-bearing mice. The saturation of the receptor protein was determined by adding the respective 100-fold excess of unlabeled PSMA-T4 (22 µg) to a bolus of [^99m^Tc]Tc-PSMA-T4 (100 μL, 0.22 µg, 10 MBq). The obtained results are presented in Figure 7 and Appendix A.

Using a 100-fold excess of unlabeled PSMA-T4 as a targeting tracer, the biodistribution data at 4 h showed a minimal (0.83 ± 0.53 %ID/g) non-specific tumor uptake, a low renal uptake (1.17 ± 0.58 %ID/g) and rapid excretion with urine.

### 2.5. Kit formulation

Pre-formulation studies were performed using PSMA-T4, which was selected based on the promising in vitro and in vivo evaluation. The influence of quantity and number of co-ligands were verified in the initial step. The radiochemical purity of [^99m^Tc]Tc-PSMA-T4 in the presence of one co-ligand ethylenediamine N,N-diacetic acid (EDDA) or tricine is presented in Appendix A.

The use of two co-ligand systems such as EDDA/tricine, and the selection of the appropriate quantity of the corresponding co-ligand to obtain radiochemical purity of PSMA-T4 over 90% is shown in Appendix A. Radiolabeling of these formulations showed radiochemical purity (RCP) >90% for 50 mg of tricine and 5 mg of EDDA as the smallest possible quantity of co-ligands necessary for properly binding technetium-99m to HYNIC chelator.

The stannous chloride dihydrate reduces technetium-99m in the form of pertechnetate to lower oxidation state. The content of stannous chloride dihydrate was tested from 10 to 150 μg/vial. It was concluded that the use of 10 μg/vial of the reducing agent resulted in satisfactory results in the labeling yield. Stannous chloride dihydrate is easily oxidized during the lyophilization process, so it is necessary to use more tin in the initial stage of the process. On the other hand, applying a higher content of stannous chloride dehydrate than 50 μg/vial increases the probability of forming undesirable impurities in the form of reduced, colloidal forms of technetium-99m. For further evaluation, the formulation containing 50 µg SnCl_2_, 2H_2_O was chosen, Appendix A.

PSMA-T4 amount should be low to avoid possible receptor saturation and undesirable pharmacological effect. 

It was shown that even 23 μg of PSMA-T4 in the labeling solutions would enable the efficient preparation of [^99m^Tc]Tc-PSMA-T4. This amount of PSMA-T4 assures repeatable achievement of the [^99m^Tc]Tc-PSMA-T4 of the highest quality and stability even using for radiolabeling technetium-99m of radioactivity as high as 1.5 GBq.

The composition of the developed PSMA-T4 kit, which allows obtaining the technetium-99m labeled PSMA inhibitor with high radiochemical purity without the additional purification, is presented in Table 9.

## 3. Materials and Methods

### 3.1. Materials

All commercially available chemicals were of analytical grade and were used without further purification. Chemicals: Tricine (N-Tris(hydroxymethyl)-methyl-glycine), EDDA (Ethylenediamine-N,N′-diacetic acid, stannous chloride dihydrate (SnCl_2_, 2H_2_O), disodium phosphatedodecahydrate (Na_2_HPO_4_, 12H_2_O), sodium dihydrogen phosphate dihydrate (NaH_2_PO_4_, 2H_2_O) were purchased from Aldrich-Sigma Chemicals Co (Poznan, Poland). Polystyrene-based Wang resin, all amino acids, and coupling reagents—N,N′-diisopropylcarboxydiimide, OxymaPure, and COMU were purchased from Iris Biotech GmbH (Marktredwitz, Germany). HBED-CC was purchased from ABX (Radeberg, Germany). The purity of amino acids and chelators was greater than 98%.

The Na^99m^TcO_4_ was obtained from the domestic ^99^Mo/^99m^Tc generator (Radioisotope Centre POLATOM, Otwock, Poland). Technetium-99 was obtained from Amersham International PLC/GE Healthcare as NH_4_^99^TcO_4_. ITLC SG was purchased from Agilent Technologies (Santa Clara, CA, USA).

### 3.2. Synthesis of PSMA-HYNIC Derivatives

The synthesis of PSMA-HYNIC compounds (including iPSMA) and PSMA-11 were started from the solid phase synthesis of the Glu(tBu)-urea-Lys-NH2 by the method described previously [29]. Further steps of synthesis where performed by standard solid phase synthesis procedures with use of 1-[(1-(cyano-2-ethoxy-2-oxoethylideneaminooxy)dimethylaminemorpholino)]uronium hexafluorophosphate (COMU) as a coupling reagent and N,N-diisopropylethylamine as a base. The final PSMA-HYNIC ligands were purified by the preparative HPLC method on the C18 reversed phase column. The identity of the compounds was confirmed by LCMS-IT-TOF and ^1^H, ^13^C, ^15^N NMR, and purity were analyzed by the HPLC method.

### 3.3. Synthesis of [^131^I]I-MIP-1095

[^131^I]I-MIP-1095, (2S)-2-({[(1S)-1-carboxy-5-({[4-[¹³¹I]iodophenyl]carbamoyl}amino) pentyl]carbamoyl}amino)pentanedioic acid), a small molecule which has a high affinity for PSMA (IC_50_, 0.3 nm) [53] was used in the in vitro study as a reference substance. It was synthesized and radiolabeled, with minor modifications, according to method described by Maresca et al. [16]. The organostannanyl intermediate was isolated and purified before radiolabeling with ^131^I. For radiolabeling 1 GBq (~45 µL) of ^131^I was used, resulting 470 MBq of final purified [^131^I]I-MIP-1095 (specific activity 156 GBq/mg) of >90% HPLC purity.

### 3.4. [^99^Tc]Tc-PSMA-T4 Complex

The radiocomplexes with long-lived technetium-99 were obtained with 2 molar excess of ^99^Tc to peptide starting from PSMA-T4 kit and using standardized labeling conditions. The kit vial was reconstituted with 385 µL of water. To this solution, 100 µL of Na^99m^TcO_4_ (100 MBq) eluate, and 15 µL of NH_4_[^99^Tc]TcO_4_ solution in water (400 µg/mL) were added. The mixture was incubated for 15 min at 100 °C. The labeled species were analyzed by LC-MS using electrospray ionization mass spectroscopy with positive ionization mode.

### 3.5. Radiolabeling of PSMA-HYNIC Derivatives

[^99m^Tc]Tc-PSMA-T1, [^99m^Tc]Tc-PSMA-T2, [^99m^Tc]Tc-PSMA-T3, [^99m^Tc]Tc-PSMA-T4 and [^99m^Tc]Tc-iPSMA (reference substance for in vivo study) were labeled in the presence of 5 mg of EDDA, 50 mg of tricine dissolved in 0.5 mL of 0.1 M phosphate buffer pH 7.4. After dissolution of reagents, 20 µg of PSMA, 20 µL of 1 mg/mL SnCl_2_, 2H_2_O solution in 0.1 M HCl, and 0.6–1 GBq of sodium pertechnetate (^99m^Tc) solution were added. The labeling mixture was incubated for 15 min at 95 °C.

### 3.6. Quality Control

The radiochemical purity was determined by the HPLC and TLC methods. HPLC analysis was performed on a Luna C18-100Å 250 mm × 4.6 mm 5 um column (Phenomenex, Torrance, CA, USA) using a Shimadzu system equipped with a Diode array detector (DAD) and online radioactivity detector connected in series. As a mobile phase water (A) and acetonitrile (B), containing 0.1% trifluoroacetic acid, in a gradient elution (0–5 min-20% B, 5–12 min-60% B, 12–15 min-20% B, 15–20 min-20% B; 1 mL/min) were used.

Instant TLC was performed using silica gel (iTLC SG—glass microfiber chromatography plates impregnated with silica gel, 2 cm × 10 cm). The amount of free [^99m^Tc]TcO_4_- (Rf = 1) was determined in methyl ethyl ketone (MEK) and ^99m^Tc -colloid species (Rf = 0) in acetonitrile:water (1:1, *v*:*v*) as mobile phase. The distribution of radioactivity on radiochromatograms was determined using Cyclone^®^Plus (PerkinElmer, Waltham, MA, USA).

LC-MS analysis was run on Shimadzu HPLC Prominence system equipped with mass spectrometer LCMS-IT-TOF system consisting of electrospray source (ESI), ion trap (IT), time of flight analyzer (TOF), and the LCMS Solution software. The samples were analyzed using the same LC analytical method for the radiochemical purity determination with a lower TFA addition of 0.05% to the elution solvents.

The conditions of mass spectrometry analysis were as follows: ionization: electrospray; ionization mode: positive; interference voltage: 5.0 kV, CDL temperature: 270 °C; heat block temperature: 270 °C; nebulizing gas flow: 1.5 L/min; mass range m/z: 200–1700 Da and ion accumulation: 10–80 ms.

### 3.7. Lipophilicity Determination

The partition coefficient logD of the radioligands were determined by shake-flask method [54,55]. The solution of 10 to 50 MBq of technetium-99m radiolabeled PSMA ligands (T1–T4) and iPSMA in 1.0 mL of phosphate-buffered saline (PBS, pH 7.4) was added to 1.0 mL of pre-saturated n-octanol solution (*n* = 3). Vials were shaken vigorously for 10 min. To achieve quantitative phase separation, the vials were centrifuged at 1600 rpm for 5 min. The radioactivity concentration in a defined volume of both the aqueous and the organic phase in six replicates each was measured in a γ-counter (Wallac Wizard 1470, PerkinElmer, USA). The partition coefficient was calculated as the logarithm of the ratio between counts per minute (cpm) measured in the n-octanol phase to the PBS phase counts.

### 3.8. Kit Preformulation Studies

The series of “wet” ^99m^Tc-labeling of PSMA-T4 was performed to optimize the quantity and concentration of reagents, temperature, and labeling time, which were then used to develop the dry kit formulation. EDDA (1–10 mg) and tricine (10–100 mg) were dissolved in various solutions of different pH (0.1 M NaOH, 1–29 mg of Na_2_HPO_4_, 12H_2_O, 1–3 mg of NaH_2_PO_4_, 2H_2_O. To determine the optimal reaction conditions, the labeling mixtures were incubated between 10 to 30 min at 80/100 °C. In a preformulating study on the radiopharmaceutical kit, EDDA and tricine were freeze-dried separately and together to optimize freeze-drying conditions, adding PSMA-T4 (23 µg) and SnCl_2_, 2H_2_O (10–100 µg) in subsequent formulations.

### 3.9. Technetium-99m Radiolabeling PSMA-T4 kit 

For technetium labeling, the PSMA-T4 kit vial was reconstituted with 1.5 mL of [^99m^Tc]TcO_4_^−^ (740–1500 MBq), and incubated for 15 min at 95 °C. After cooling to room temperature, the reaction mixture was analyzed by HPLC and TLC methods.

### 3.10. In Vitro Study

#### 3.10.1. Cell Culture

The PSMA-positive LNCaP (ATCC, CRL-1740) cells were grown in RPMI 1640 (ATCC modified, Gibco, Waltham, MA, USA) and the PSMA-negative PC-3 (ATCC, CRL-1435) cells were grown in Nutrition Mixture F-12 (Gibco). Both media were supplemented with 10% fetal bovine serum (FBS) and 1% penicillin/streptomycin (10,000 U/mL) (Merck, Kenilworth, NJ, USA). The cultures were maintained at 37 °C in humidified 5% CO_2_ atmosphere. The cells were harvested using trypsin-EDTA solution (0.25% trypsin, 0.02% EDTA, Merck) after reaching 80% confluence.

#### 3.10.2. Cell Uptake

The LNCaP cells in concentration 5 × 10^5^ per well were cultured on 12 wells plates to confluence. After 48 h of incubation, the wells were washed with un-supplemented RPMI 1640. The cells were incubated with one concentration of radiolabeled compounds for 2 h. After incubation, cells were washed with PBS. The binding of tested radiocomplexes to PSMA antigen was evaluated by measurement of the radioactivity of 1 mL of glycine buffer (50 mM in 0.1 M NaCl, pH 2.8) used for rinsing the cells. The internalization was determined by radioactivity of 1 mL 1 M NaOH used for the lysis. Total binding was calculated as a sum of membrane and internalized fraction.

#### 3.10.3. Cell Membrane Isolation

Cells from prostate cancer carcinomas: LNCaP and PC3, were isolated like previously described [56]. The cells were harvested using 1 mL/flask of Trypsin-EDTA solution and centrifuged (1500 rpm for 15 min (LNCaP) and 1300 rpm for 6 min (PC3)). Received cell sediment was resuspended in 9.7 mM NaHCO_3_ containing 0.3 mM NaH_2_PO_4_ and incubated for 30 min at 4 °C. The solution was dispensed on 1.5 mL Eppendorf and centrifuged (13,500 rpm for 30 min). The supernatant was removed, the pellet was resuspended in 50 mM Tris-HCl Buffer (pH 7.4) and homogenized using a syringe with a thin needle. The solution was centrifuged (13,500 rpm for 30 min), and the final pellet was suspended in 50 mM Tris-HCl buffer (pH 7.4) and transferred to −70 °C.

#### 3.10.4. Saturation Binding

The K_d_ value for ^99m^Tc-labeled PSMA ligands were determined by saturation binding analysis on membranes isolated from LNCaP (PSMA+). Unspecific binding was evaluated using PC3 (PSMA−) cell membrane, and specific binding was evaluated as a difference of total binding (LNCaP) and unspecific binding to the human prostate cancer cell line (PC3). Cells were incubated with 41–131,000 pM radiolabeled PSMA inhibitors for 2 h at 37 °C on special MultiScreen™ 96 well assay plates (Merck). The supernatant solution was filtered under vacuum, and the membranes were washed twice using phosphate buffer saline (PBS, IITD Wroclaw). The filters containing membranes with the attached radiotracer were extruded into tubes using MultiScreen Multiple Punch (Merck). The number of associated inhibitors was determined by measuring the radioactivity of the filters using Wallac Wizard 1470 γ counter. The K_d_ was analyzed and calculated by nonlinear regression using GraphPad Prism 9.4 statistic program.

#### 3.10.5. Competitive Binding

The LNCaP cell membranes were distributed (in four replicates) in the constant amount on special MultiScreen™ 96 well assay plates. 100 µL of un-supplemented RPMI 1640 medium and 50 µL of increasing concentration of the unlabeled PSMA inhibitors was added to each well. As a control, 50 µL of PBS in 100 µL of media was used. After 15 min incubation at 37 °C, 50 µL of [^131^I]I-MIP-1095 was added. Plates were incubated for 2 h at 37 °C. After an appropriate time, the supernatant solution was filtered under vacuum, and the membranes were washed twice with PBS solution. Filters and membranes were extruded to the tubes using MultiScreen Multiple Punch. The radioactivity of the filters was measured using a Wallac Wizard 1470 γ counter. The IC50 was evaluated and determined using GraphPad Prism 9.4 statistic program. As a control, PSMA-11 was used.

### 3.11. In Vivo Study

#### 3.11.1. Animal Models

BALB/c NUDE male mice (5–6 weeks old, mean body mass of 20 g) were purchased from the Charles River Laboratories (Sulzfeld, Germany). Wistar male rats and BALB/c mice (both 5–7 weeks old) were purchased from the M. Mossakowski Institute of Experimental and Clinical Medicine, Polish Academy of Sciences in Warsaw (Poland).

On arrival, animals were housed for 5 days in groups of five in standard cages (Wistar rats and BALB/c mice) and IVS cages (BALB/c NUDE) in the animal facility of the Radioisotope Centre POLATOM (Otwock, Poland). They were housed in a quiet room under constant conditions (22 °C, 50% relative humidity, 12-h light/dark cycles with dark periods from 7 p.m. to 7 a.m.) with free access to standard food and water. Veterinarian staff and investigators observed the rodents daily to ensure animal welfare and determine if humane endpoints were reached (e.g., hunched and ruffled appearance, apathy, ulceration, severe weight loss, tumor burden). Experimental procedures were carried out in conformity with the National Legislation and the Council Directive of the European Communities on the Protection of Animals Used for Experimental and Other Scientific Purposes (2010/63/UE) and the “ARRIVE guidelines for reporting animal research” [57]. The POLATOM protocol was approved by the Ist Local Animal Ethics Committee in Warsaw (authorization 681/2018, approval date 4 July 2018 for both type of mice, and authorization 400/2017, approval date 21 November 2017 for rats).

The standard protocol involved animals randomized into fixed groups (five per group). Before injection, the labeled compounds were diluted in 0.9% NaCl and then intravenously injected (0.1 mL per mice and 0.2 mL per rat). At established time points after-injection, the animals were euthanized by cervical dislocation and dissected. Selected organs and tissues were weighed and their radioactivity was measured in a gamma counter equipped with a NaI(Tl) crystal. The physiological distribution was calculated and expressed in terms of the percentage of administrated radioactivity found in each of the selected organs or tissues per gram (%ID/g) with the aid of suitable standards of the injected dose.

#### 3.11.2. Physiological Distribution in Healthy and Xenografted Animals

The physiological distribution of four radioligands: [^99m^Tc]Tc-PSMA-T1, [^99m^Tc]Tc-PSMA-T2, [^99m^Tc]Tc-PSMA-T3 and [^99m^Tc]Tc-PSMA-T4 [^99m^Tc]Tc-iPSMA was performed at a one-time point (4 h i.v.) For the experiment, healthy BALB/c (*n* = 5) mice were injected with a dose of 0.44 µg PSMA in 0.1 mL and activity of 9.8 MBq.

The pharmacokinetic analysis of the radiolabeled [^99m^Tc]Tc-PSMA-T4 (0.2 µg, 0.2 mL, 9.8 MBq) was investigated at eight-time points (15 min, 30 min, 1 h, 2 h, 4 h, 6 h, 15 h and 24 h) in groups of five rats per group. The pharmacokinetic parameters were determined according to a one-phase exponential decay model. 

The xenografted mice were prepared according to the methods previously given in the literature [29]. The animals were randomized into two groups (five mice per group): Group 1, treated with a single intravenous injection of [^99m^Tc]Tc-PSMA-T4 (0.1 mL, 0.2 µg, 9.5 MBq, SA 16.7 GBq/mg = 17.2 GBq/µmol) and Group 2, where the PSMA receptors were locally blocked with a single intravenous co-injection of PSMA-T4 (dose range: 100× mass equivalent of the radioactive [^99m^Tc]Tc-PSMA-T4 dose). The mice were euthanized after 4 h p.i.v.

### 3.12. Statistics

Results are provided as mean ± SD.

The results of physiological distribution expressed as a percentage of the dose administered per gram of tissue (%ID/g) were presented in the form of an average with standard deviation (mean ± standard deviation (SD)), with *n* representing the number of animals per group. Data were statistically analyzed using GraphPad Prism version 9.4 for Windows and tested for normal distribution with the Kolmogorov–Smirnov test. In case of normal distribution, results were assessed by two-tailed, unpaired Student’s *t*-tests. Otherwise, results were assessed by two-way ANOVA. A *p* value of <0.05 with two-tailed testing was considered statistically significant.

For blood activity data in healthy Wistar rats, a mono-exponential decay model (Equation (1)) was used to describe the percentage of remaining activity (%ID/g) as a function of time post-injection (t):%ID/g = Span ∗ exp(−K ∗ t) + Plateau,(1)
where:Span is the difference between %ID/g (0) and Plateau%ID/g (0) is the %ID/g value when t (time) is zeroPlateau is the %ID/g value at infinite timesK is the rate constants

For renal excretion of [^99m^Tc]Tc-PSMA-T4, the two-phases model was applied (Equation (2)).
%ID/g = Span1 ∗ exp(−K1 ∗ t) + Span2 ∗ exp(-K2 ∗ t) Plateau,(2)
where:Span1 = (%ID/g (0)-Plateau) ∗PercentFast∗0.01Span2 = (%ID/g (0)-Plateau) ∗ (100-PercentFast) ∗0.01%ID/g (0)—the %ID/g value when time is zeroPlateau—the %ID/g value at infinite timesK is the rate constant

Nonlinear least-squares regression was used to estimate the half-life of the exponential functions:T_1/2_ = ln(2)/K,(3)

## 4. Conclusions

This work presents the development and preclinical evaluation of a novel PSMA-T4 ligand (Glu-CO-Lys-L-Trp-4-Amc-HYNIC), which showed excellent radiolabeling characteristics, high selectivity towards PSMA receptors in vitro and favorable tumor accumulation in LNCaP tumor-bearing mice. As a final result, we prepared a single vial lyophilized kit to prepare [^99m^Tc]Tc-PSMA-T4 with desired radiochemical and pharmaceutical purity. This kit allows for rapid and reproducible preparation of radiopharmaceuticals in a hospital environment [58]. Preclinical studies in the tumor-bearing mice indicated a high tracer accumulation in the PSMA-positive tumor and led to steadily increasing tumor to muscle ratios (T/M) over time (e.g., T/M: 78, 174, 262 after 2 h, 6 h, and 24 h p.i.v, respectively). Such properties mean that [^99m^Tc]Tc-PSMA-T4 can be an effective tracer for SPECT imaging, even after a long time after administration. It may therefore also prove useful for the radio-guided surgery (RGS) of patients with mPCa. Furthermore, [^99m^Tc]Tc-PSMA-T4 showed effectiveness in SPECT studies in humans, and soon, its usefulness will be extensively evaluated in phase 2/3 clinical trials (EudraCT No. 2021-005113-14).

## 5. Patents

The work reported in this manuscript results as a patents: PL239934B1 [39]—Pochodne inhibitorów PSMA do znakowania ^99m^Tc poprzez HYNIC, zestaw radiofarmaceutyczny, preparaty radiofarmaceutyczne oraz ich zastosowanie w diagnostyce raka prostaty,; available on line: https://patents.google.com/patent/PL239934B1/en (accessed on 7 September 2022); EP3721907A1 [38]—PSMA inhibitor derivatives for labelling with ^99m^Tc via HYNIC, A radiopharmaceutical kit, radiopharmaceutical preparation and their use in prostate cancer diagnostics.; available online: https://worldwide.espacenet.com (accessed on 7 September 2022), US 11426395 B2 [59]—PSMA inhibitor derivatives for labelling with ^99m^Tc via HYNIC, A radiopharmaceutical kit, radiopharmaceutical preparation and their use in prostate cancer diagnostics.; available online: https://worldwide.espacenet.com (accessed on 7 September 2022).

## Figures and Tables

**Figure 1 molecules-27-07216-f001:**
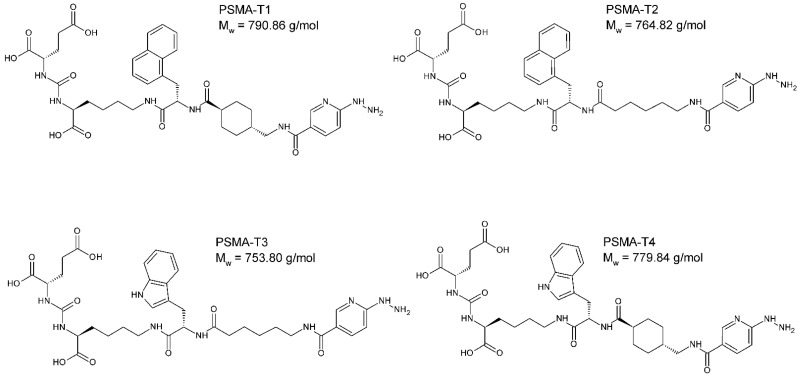
The chemical structure of PSMA ligands: PSMA-T1: Glu-CO-Lys-L2NaI-6Ahx-HYNIC, PSMA-T2: Glu-CO-Lys-L2NaI-4Amc-HYNIC, PSMA-T3: Glu-CO-Lys-LTrp-6Ahx-HYNIC, PSMA-T4: Glu-CO-Lys-LTrp-4Amc-HYNIC, (S)-2-{3-[(S)-5-[(S)-3-(1H-Indol-3-yl)-2-({(1r,4r)-4-[(6-hydrazino-nicotinoylamino)methyl]cyclohexyl}carbonylamino)propionylamino]-1-carboxypentyl] ureido}glutaric acid.

**Figure 2 molecules-27-07216-f002:**
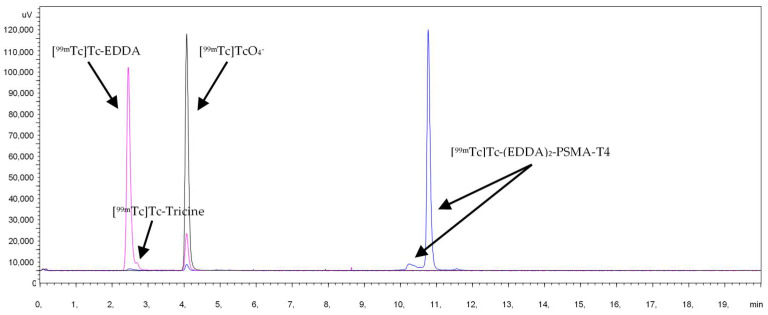
HPLC chromatogram of [^99m^Tc]Tc-(EDDA)_2_-HYNIC-PSMA-T4 and radiochemical impurities.

**Figure 3 molecules-27-07216-f003:**
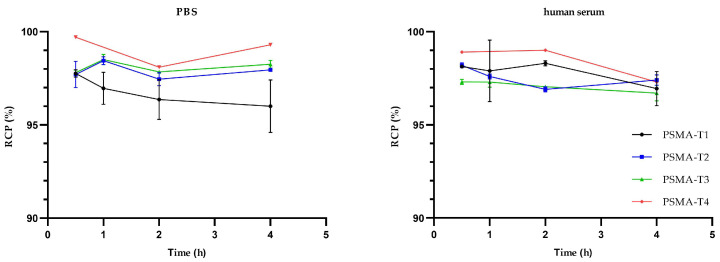
Stability of ^99m^Tc-labelled PSMA-HYNIC compounds in PBS and human serum at 37 °C.

**Figure 4 molecules-27-07216-f004:**
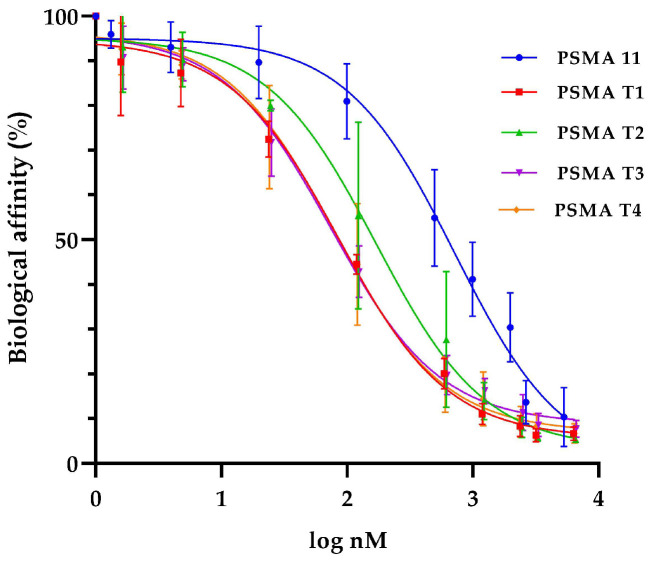
Binding affinity for tested PSMA derivatives on LNCaP prostate cell membranes and [^131^I]I-MIP1095 as a competitive radioligand.

**Figure 5 molecules-27-07216-f005:**
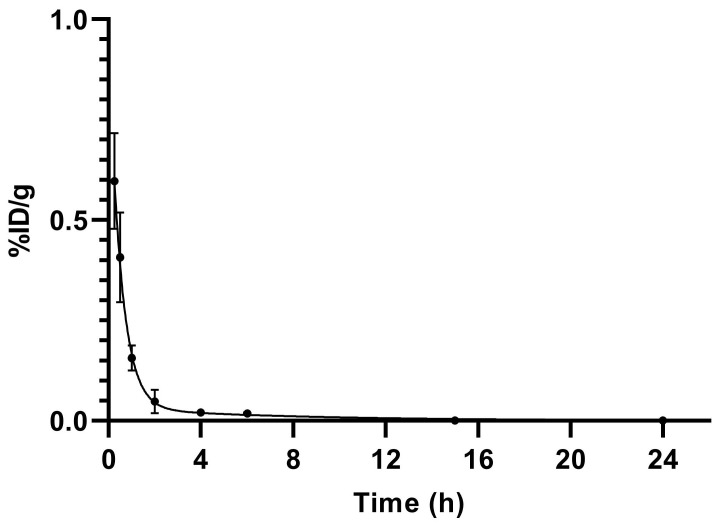
Blood elimination of [^99m^Tc]Tc-PSMA-T4 (*n* = 5) in healthy male Wistar rats. Data presented as a mean and standard deviation.

**Figure 6 molecules-27-07216-f006:**
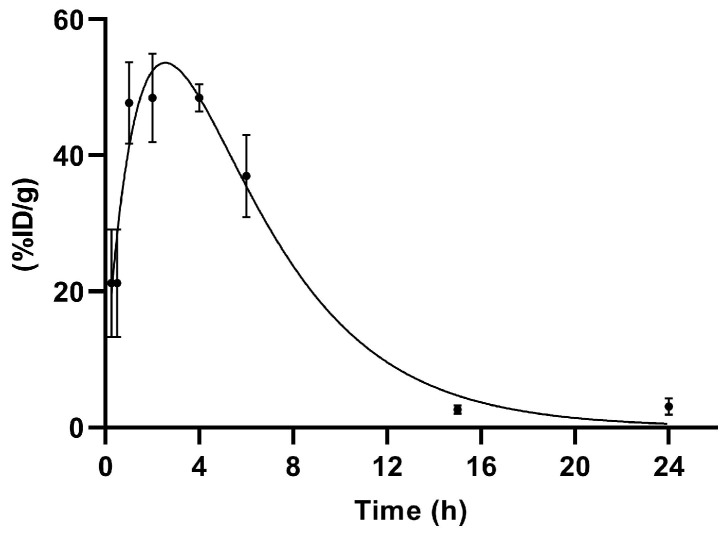
Renal excretion of [^99m^Tc]Tc-PSMA-T4 (*n* = 5) in healthy male Wistar rats.

**Figure 7 molecules-27-07216-f007:**
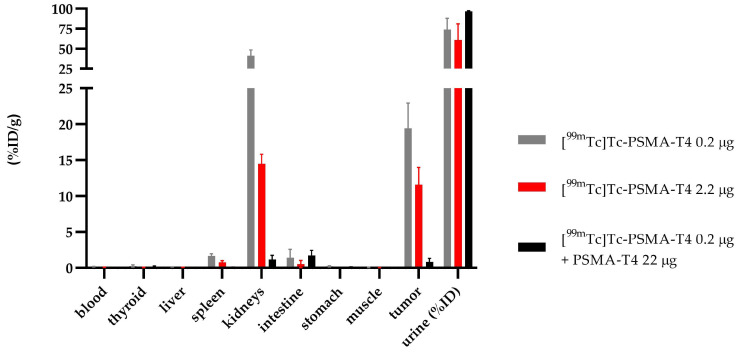
Biodistribution of [^99m^Tc]Tc-PSMA-T4 in BALB/c Nude injected in two doses of radiolabeled PSMA-T4 (0.22 µg and 2.2 µg, gray and red bar, respectively) and as co-injection with an excess of unlabeled PSMA (black bar).

**Table 1 molecules-27-07216-t001:** LC-MS data for [^99/99m^Tc]Tc-PSMA-T4 complex.

Rt (min)	m/z Observed	Assignment	m/z Calculated
9.9	613.64	[(PSMA-T4 + Tc + 2EDDA - 5H) + 2H]^2+^	613.78
	1226.27	[(PSMA-T4 + Tc + 2EDDA - 5H) + H]^+^	1226.5
10.2	613.64	[(PSMA-T4 + Tc + 2EDDA - 5H) + 2H]^2+^	613.78
	1226.28	[(PSMA-T4 + Tc + 2EDDA - 5H) + H]^+^	1226.56

**Table 2 molecules-27-07216-t002:** Comparison of lipophilicity of PSMA radiocomplexes.

Compound	logD
[^99m^Tc]Tc-PSMA-T1	−3.81 ± 0.01
[^99m^Tc]Tc-PSMA-T2	−3.43 ± 0.02
[^99m^Tc]Tc-PSMA-T3	−4.88 ± 0.05
[^99m^Tc]Tc-PSMA-T4	−4.76 ± 0.06
[^99m^Tc]Tc-iPSMA	−3.42 ± 0.01

**Table 3 molecules-27-07216-t003:** Binding, internalization, and cell uptake of PSMA inhibitors.

PSMA Inhibitor	Binding (%)	Internalization (%)	Cell Uptake (%)
[^99m^Tc]Tc-PSMA-T1	24.0	7.5	31.5
[^99m^Tc]Tc-PSMA-T2	17.8	9.7	27.5
[^99m^Tc]Tc-PSMA-T3	18.5	9.3	27.8
[^99m^Tc]Tc-PSMA-T4	19.2	8.3	27.5

**Table 4 molecules-27-07216-t004:** The K_d_ values and specific binding of the PSMA inhibitors were determined in the saturation binding assay.

PSMA inhibitor	K_d_ (nM)	Specific Binding (%)
[^99m^Tc]Tc-PSMA-T1	10.2 ± 1.9	99.0 ± 0.6
[^99m^Tc]Tc-PSMA-T2	9.2 ± 2.4	99.1 ± 0.4
[^99m^Tc]Tc-PSMA-T3	6.1 ± 1.6	99.8 ± 0.1
[^99m^Tc]Tc-PSMA-T4	5.4 ± 2.3	98.0 ± 0.1
[^68^Ga]Ga-PSMA-11	11.4 ± 7.1 [15]	-

**Table 5 molecules-27-07216-t005:** IC50 values for tested PSMA derivatives on LNCaP prostate cell membranes and [^131^I]I-MIP1095 as a competitive radioligand.

	PSMA-T1	PSMA-T2	PSMA-T3	PSMA-T4	PSMA-11
IC50 (nM)	85.8	168.9	74.6	79.5	714.9

**Table 6 molecules-27-07216-t006:** Results of biodistribution study of ^99m^Tc-PSMA ligands in BALB/c mice (*n* = 5) at 4 h after i.v.; (%ID/g. mean ± SD).

	[^99m^Tc]Tc-iPSMA	[^99m^Tc]Tc-PSMA-T1	[^99m^Tc]Tc-PSMA-T2	[^99m^Tc]Tc-PSMA-T3	[^99m^Tc]Tc-PSMA-T4
blood	0.13 ± 0.02	0.21 ± 0.02	0.44 ± 0.20	0.08 ± 0.02	0.10 ± 0.04
thyroid	0.15 ± 0.09	0.94 ± 0.60	2.74 ± 1.22	0.18 ± 0.13	0.20 ± 0.13
heart	0.18 ± 0.05	0.19 ± 0.11	0.27 ± 0.02	0.08 ± 0.02	0.09 ± 0.06
lung	0.24 ± 0.04	0.33 ± 0.05	0.37 ± 0.08	0.11 ± 0.03	0.19 ± 0.04
liver	0.18 ± 0.02	0.17 ± 0.02	0.81 ± 0.08	0.17 ± 0.11	0.08 ± 0.01
spleen	1.06 ± 0.36	2.64 ± 1.09	1.80 ± 0.39	0.48 ± 025	0.67 ± 0.13
pancreas	0.24 ± 0.12	0.61 ± 0.16	0.53 ± 0.28	0.14 ± 0.07	0.09 ± 0.04
kidney	28.12 ± 2.52	68.49 ± 7.64	78.10 ± 16.25	19.23 ± 3.95	15.90 ± 2.15
intestine	0.88 ± 0.03	1.13 ± 0.68	2.58 ± 0.48	0.96 ± 0.45	0.65 ± 0.52
stomach wall	0.16 ± 0.07	0.25 ± 0.04	1.19 ± 0.26	0.17 ± 0.09	0.15 ± 013
femur	0.28 ± 0.09	0.24 ± 0.10	0.34 ± 0.02	0.14 ± 0.13	0.11 ± 0.05
muscle	0.10 ± 0.04	0.11 ± 0.07	0.14 ± 0.08	0.03 ± 0.03	0.04 ± 0.03
urine (%ID)	84.96 ± 1.43	69.74 ± 1.41	59.21 ± 4.30	88.12 ± 3.35	91.23 ± 1.09

**Table 7 molecules-27-07216-t007:** Results of pharmacokinetics study of [^99m^Tc]Tc-PSMA-T4 complex in Wistar rats (*n* = 5) after intravenous administration (%ID/g, mean ± SD).

	**15 min.**	**30 min.**	**1 h**	**2 h**
blood	0.59 ± 0.12	0.41 ± 0.11	0.15 ± 0.03	0.05 ± 0.03
thyroid	1.71 ± 0.45	0.90 ± 0.78	0.18 ± 0.06	0.12 ± 0.06
liver	0.40 ± 0.09	0.35 ± 0.04	0.26 ± 0.05	0.13 ± 0.04
spleen	0.22 ± 0.06	0.17 ± 0.04	0.14 ± 0.10	0.07 ± 0.02
kidney	21.22 ± 7.92	24.37 ± 0.52	47.73 ± 6.00	48.49 ± 6.50
intestine	0.20 ± 0.05	0.14 ± 0.04	0.08 ± 0.02	0.06 ± 0.02
stomach wall	0.51 ± 0.23	0.50 ± 0.08	0.12 ± 0.03	0.06 ± 0.02
femur	0.20 ± 0.07	0.15 ± 0.05	0.07 ± 0.02	0.04 ± 0.01
muscle	0.14 ± 0.04	0.09 ± 0.03	0.06 ± 0.01	0.03 ± 0.01
salivary gland	0.43 ± 0.12	0.32 ± 0.07	0.27 ± 0.08	0.14 ± 0.03
urine (%ID)	0.61 ± 0.393	1.30 ± 0.82	1.57 ± 0.47	2.01 ± 1.14
	**4 h**	**6 h**	**15 h**	**24 h**
blood	0.02 ± 0.00	0.02 ± 0.00	0.00 ± 0.00	0.00 ± 0.00
thyroid	0.04 ± 0.04	0.06 ± 0.05	0.01 ± 0.02	0.00 ± 0.00
liver	0.06 ± 0.01	0.05 ± 0.04	0.01 ± 0.00	0.01 ± 0.00
spleen	0.03 ± 0.01	0.04 ± 0.01	0.02 ± 0.03	0.01 ± 0.00
kidney	48.48 ± 2.02	36.98 ± 6.07	2.62 ± 0.60	3.06 ± 1.21
intestine	0.04 ± 0.01	0.05 ± 0.02	0.07 ± 0.00	0.09 ± 0.05
stomach wall	0.03 ± 0.00	0.03 ± 0.01	0.02 ± 0.01	0.01 ± 0.00
femur	0.02 ± 0.01	0.03 ± 0.01	0.01 ± 0.01	0.00 ± 0.01
muscle	0.01 ± 0.00	0.01 ± 0.01	0.00 ± 0.01	0.00 ± 0.01
salivary gland	0.06 ± 0.02	0.05 ± 0.02	0.01 ± 0.02	0.00 ± 0.00
urine (%ID)	14.99 ± 3.16	38.96 ± 6.40	81.12 ± 5.46	81.11 ± 2.51

**Table 8 molecules-27-07216-t008:** Results of biodistribution of the [^99m^Tc]Tc-PSMA-T4 complex in BALB/c NUDE mice bearing LNCaP tumor (%ID/g, mean ± SD, *n* = 4).

	1 h	2 h	4 h	6 h	24 h
blood	0.81 ± 0.09	0.34 ± 0.09	0.13 ± 0.03	0.15 ± 0.08	0.10 ± 0.05
thyroid	0.41 ± 0.16	0.14 ± 0.07	0.20 ± 0.09	0.09 ± 0.03	0.05 ± 0.09
liver	0.43 ± 0.05	0.18 ± 0.04	0.12 ± 0.01	0.08 ± 0.01	0.09 ± 0.04
spleen	6.07 ± 1.44	4.10 ± 0.92	1.67 ± 0.31	1.46 ± 0.46	0.30 ± 0.17
kidney	141.08 ± 25.79	132.72 ± 23.37	41.22 ± 7.50	41.51 ± 5.56	7.16 ± 3.17
small intestine	0.62 ± 0.15	0.28 ± 0.06	0.24 ± 0.03	0.18 ± 0.06	0.17 ± 0.06
large intestine	0.30 ± 0.02	0.12 ± 0.04	0.66 ± 0.27	0.72 ± 0.63	1.64 ± 1.03
stomach wall	0.50 ± 0.13	0.25 ± 0.04	0.20 ± 0.07	0.09 ± 0.03	0.27 ± 0.14
muscle	0.32 ± 0.04	0.11 ± 0.03	0.06 ± 0.01	0.06 ± 0.02	0.10 ± 0.06
tumor	10.09 ± 5.62	26.45 ± 1.42	20.77 ± 3.99	26.12 ± 9.11	26.21 ± 12.09
urine (%ID)	23.08 ± 4.76	41.69 ± 11.26	80.09 ± 3.78	73.00 ± 12.53	90.36 ± 4.87

**Table 9 molecules-27-07216-t009:** Final kit composition of the PSMA-T4 for radiopharmaceutical preparation.

Substance	Amount per Vial	Function
PSMA-T4	23 µg	Active substance
Tricine	50 mg	Co-ligand
EDDA	5 mg	Co-ligand
SnCl_2_, 2H_2_O	50 µg	Reducing agent
Na_2_HPO_4_, 12H_2_O	29 mg	Excipient for pH adjustment
NaH_2_PO_4_, 2H_2_O	3 mg	Excipient for pH adjustment
Nitrogen	q.s.	Protective gas

## Data Availability

The data presented in this study are available on request from the corresponding authors.

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
