# Peer review of "[^99m^Tc]Tc-PSMA-T4—Novel SPECT Tracer for Metastatic PCa: From Bench to Clinic"

_molecules, 2022, doi:10.3390/molecules27217216_

Round 1
Reviewer 1 Report
Title: “[99mTc]Tc-PSMA-T4 – novel SPECT tracer for metastatic PCa: from bench to clinic”
Authors: Michał Maurin, Monika Wyczółkowska, Agnieszka Sawicka, Arkadiusz Sikora, Urszula Karczmarczyk, Barbara Janota, Marcin Radzik, Dominik Kłudkiewicz, Justyna Pijarowska-Kruszyna, Antoni Jaroń, Wioletta Wojdowska, Piotr Garnuszek
Affiliation: National Centre for Nuclear Research, Radioisotope Centre POLATOM, 05-400 Otwock, Poland.
Major Concerns: Potential bias and conflicts of interest: The authors have a patent on a HYNIC chelating system, [99mTc]Tc-PSMA-T4, a preparation developed by their team [references 38, 39]. Despite this the authors claim no potential or conceived conflicts of interest. “Conflicts of Interest: The authors have no conflicts of interest to declare.” Page 17, line 615.
Abstract: Is concise
Introduction: Provides adequate background information.
Results and Discussion: Well presented.
Materials and Methods: Adequate for replication.
Conclusion: Very limited discussion of the potential benefits of 99Tc SPECT imaging for PSMA over PET imaging and timing of imaging for PSMA after renal clearance of free 99Tc and metabolized [99mTc]Tc-PSMA-T4 based on their experimental observations.
Line by line critique:
Page 2, Line 62; “…is enhanced in poorly differentiated, metastatic and hormone-refractory carcinomas, at hence it can be a valuable target for developing radiopharmaceutical and radiolabeled …” Typo? -Suggest ‘…is enhanced in poorly differentiated, metastatic and hormone-refractory carcinomas, and hence it can be a valuable target for developing radiopharmaceutical and radiolabeled …’
Page 2, Line 64; “…such as urea-based PSMA inhibitors, for precisely diagnosis, staging and treatment of PCa …” Syntax – suggest ‘…such as urea-based PSMA inhibitors, for precisely diagnosing, staging and treating PCa…’
Page 2, Line 67; “…have great potential in preclinical and clinical stages, enabling effective diagnosis and therapy of prostate cancer.” Preference – Suggest continuing use of abbreviations once introduced. ‘…have great potential in preclinical and clinical stages, enabling effective diagnosis and therapy of PCa.’
Page 3, line 129; “…compounds was confirmed the mass spectroscopy (MS) and, in the case of PSMA-T4, additionally by the nuclear magnetic resonance (NMR),…” Typo – suggest ‘…compounds was confirmed by mass spectroscopy (MS) and, in the case of PSMA-T4, additionally by the nuclear magnetic resonance (NMR),…’
Page 5, line 176; “There is no direct possibility to to identify which EDDA donor atoms bind to technetium. Typo -Suggest “There is no direct possibility to identify which EDDA donor atoms bind to technetium.’
Page 5, line 180; “…stability in phosphate buffer saline (PBS) and human serum at 37°C by TLC method revealed to be stable for at least 4h, Figure 3 with no…” Syntax – Suggest ‘…stability in phosphate buffer saline (PBS) and human serum at 37°C by TLC method revealed them to be stable for at least 4h, Figure 3 with no…’
Page 12, line 349; “Technetium-99 material was obtained from Amersham International plc as NH4 99TcO4.” Correction- Suggest ‘Technetium-99 material was obtained from Amersham International PLC/ GE Healthcare as NH4 99TcO4.’
Page 13, line 388; “…Shimadzu system equipped with a DAD…” spell out abbreviations in first instance – suggest ‘…Shimadzu system equipped with a Diode array detector (DAD)…’
Page 13, line 410; “Vials were shacked vigorously…” Typo – suggest ‘Vials were shaken vigorously…’
Page 14, line 467; “… MultiScreen Multile Punch (Merck).” Typo – suggest ‘…MultiScreen Multiple Punch (Merck).’
Author Response
Responses to Reviewer #1 comments
Major Concerns: Potential bias and conflicts of interest: The authors have a patent on a HYNIC chelating system, [99mTc]Tc-PSMA-T4, a preparation developed by their team [references 38, 39]. Despite this the authors claim no potential or conceived conflicts of interest. “Conflicts of Interest: The authors have no conflicts of interest to declare.” Page 17, line 615.
Indeed, the issue of conflict of interest raised by reviewer #1 escaped our attention. In the revised version of the manuscript, we disclosed that authors: M.M., M.W., A.E.S., U.K., B.J., M.R., J.P-K., A.J. and P.G are the inventor of the patents: PL239934B1 and US 11426395 B2.
Conclusion: Very limited discussion of the potential benefits of 99Tc SPECT imaging for PSMA over PET imaging and timing of imaging for PSMA after renal clearance of free 99Tc and metabolized [99mTc]Tc-PSMA-T4 based on their experimental observations.
To further infer the potential diagnostic efficacy, information on the physiological distribution of the preparation in preclinical studies in tumour-bearing mice was inserted in Conclusion.
"Preclinical studies in the tumour-bearing mice indicated high tracer accumulation in the PSMA-positive tumour and led to steadily increasing tumour to muscle ratios (T/M) over time (e.g. T/M: 78, 174, 262 after 2h, 6h, and 24h p.i.v, respectively). Such properties mean that [99mTc]Tc-PSMA-T4 can be an effective tracer for SPECT imaging, even after a long time after administration. It may therefore also prove useful for the radioguided surgery (RGS) of patients with mPCa."
We thank the Reviewer for pointing out linguistic errors in the text. We have corrected the text as suggested.
Please find in attachment the corrected manuscript.
Reviewer 2 Report
Dear authors,
After reviewing this study, I have two concerns:
30%-75% metastatic site of prostate cancer is bone marrow. Did the authors test the biodistribution of tested compounds in bone marrow?
The authors exhibited a stable radiolabel activity of the test compounds in tumor over 24 h. How about the half-life of the test compound on the tumor?
From table 8, we can find that most test compound are absorbed by kidney and subsequently excreted from urine. In my opinion, the Infused dosage of test compound maybe overdosed. I suggest decreasing the dosage to find the optimal dosage for the compound.
Author Response
Responses to Reviewer #2 comments
30%-75% metastatic site of prostate cancer is bone marrow. Did the authors test the biodistribution of tested compounds in bone marrow?
The biodistribution of the tested complexes in bone marrow was not studied.
The LNCaP cells used for in vivo experiments form solid subcutaneous tumours. Due to fact that these cells are not metastatic for bone marrow (LNCaP clone FGC - CRL-1740 | ATCC) , such experiments where not justified. Additionally in the pharmacokinetic study, the uptake in bone was assessed by measurement of accumulation of the tracer in the femur. It was observed that at all time points tested, the accumulation in the femur was marginal, decreasing in time (Table 7: Results of pharmacokinetics study of [99mTc]Tc-PSMA-T4 complex in Wistar rats (n=5) after intravenous administration (%ID/g, mean±SD). These observations indicate no significant and specific accumulation of the tracer in the bone marrow.
The authors exhibited a stable radiolabel activity of the test compounds in tumour over 24 h. How about the half-life of the test compound on the tumor?
We observed a stable accumulation of the radiotracers in the tumour (over 24h, Table 8: Results of biodistribution of the [99mTc]Tc-PSMA-T4 complex in BALB/c NUDE mice bearing LNCaP tumour (%ID/g, mean±SD, n = 4). The developed tracer is intended as a diagnostic tracer, labelled with short-lived 99mTc (~6h half-life). In that case, the optimal time for patient imaging is 2-6h after administration of the tracer. However, due to the persistent accumulation of the compound in the tumour and short half-life of the technetium-99m the half-life of the PSMA-T4 in the tumour was not tested. However, considering the constant percentage of radioactivity in the tumour over a time interval of 2-24 h, it can be assumed that the effective half-life of the radiolabel in the tumour is equal to the physical half-life of technetium-99m.
From table 8, we can find that most test compound are absorbed by kidney and subsequently excreted from urine. In my opinion, the Infused dosage of test compound maybe overdosed. I suggest decreasing the dosage to find the optimal dosage for the compound.
Please refer to the results of the biodistribution of 99mTc-PSMA-T4 at three doses (Figure 7: Biodistribution of [99mTc]Tc-PSMA-T4 in BALB/c Nude injected in two doses of radiolabelled PSMA-T4 (0.22µg and 2.2 µg, grey and red bar, respectively) and as co-injection with an excess of un-labelled PSMA-T4 (black bar). In case of lower doses we observed favourable accumulation of radioactivity in tumour. The accumulation in tumour and kidneys decreases with increasing PSMA-T4 dose. It can be explained by the saturation of receptors in the tumour and kidneys with a “cold” tracer in higher doses, resulting in a lower accumulation of radiotracer.
In case of the in vivo study in small animals the limiting parameter is the radioactivity of the tracer that can be measured with acceptable error. In case of applied dose (0.2 mcg / animal) the radioactivity administered was ~5MBq per animal. Lower radioactivity would result with uncertainty of radioactivity measurements. Because of that, the smaller doses were not tested.